# Research on the Recognition of Various Muscle Fatigue States in Resistance Strength Training

**DOI:** 10.3390/healthcare10112292

**Published:** 2022-11-15

**Authors:** Yinghao Wang, Chunfu Lu, Mingyu Zhang, Jianfeng Wu, Zhichuan Tang

**Affiliations:** Industrial Design Department, Zhejiang University of Technology, Hangzhou 310023, China

**Keywords:** resistance strength training, dynamic muscle fatigue, recognition, sEMG signal, convolutional neural network

## Abstract

Instantly and accurately identifying the state of dynamic muscle fatigue in resistance training can help fitness trainers to build a more scientific and reasonable training program. By investigating the isokinetic flexion and extension strength training of the knee joint, this paper tried to extract surface electromyogram (sEMG) features and establish recognition models to classify muscle states of the target muscles in the isokinetic strength training of the knee joint. First, an experiment was carried out to collect the sEMG signals of the target muscles. Second, two nonlinear dynamic indexes, wavelet packet entropy (WPE) and power spectrum entropy (PSE), were extracted from the obtained sEMG signals to verify the feasibility of characterizing muscle fatigue. Third, a convolutional neural network (CNN) recognition model was constructed and trained with the obtained sEMG experimental data to enable the extraction and recognition of EMG deep features. Finally, the CNN recognition model was compared with multiple support vector machines (Multi-SVM) and multiple linear discriminant analysis (Multi-LDA). The results showed that the CNN model had a better classification accuracy. The overall recognition accuracy of the CNN model applied to the test data (91.38%) was higher than that of the other two models, which verified that the CNN dynamic fatigue recognition model based on subjective and objective information feedback had better recognition performance. Furthermore, training on a larger dataset could further improve the recognition accuracy of the CNN recognition model.

## 1. Introduction

Among existing fitness training methods, resistance strength training, also known as “anaerobic training”, is the most effective training method for muscle building and shaping [1,2]. Isokinetic exercise is one type of resistance strength training. The joints of the human body keep rotating at a constant speed during the whole exercise process, and the exercise resistance changes with the change in the muscle force output. It is recognized as the most advanced muscle strength training method, and it is considered a relatively safe and proper strength training method [3,4,5]. However, there are several problems when ordinary people undergo isokinetic strength training: (1) insufficient training, resulting in poor training effects; (2) excessive training, which may cause muscle damage. Exercise physiology research has shown that in resistance training, it is necessary to evaluate the initial fatigue state of the muscle, and then set the corresponding scientific exercise load and volume, so as to train the muscle to the extreme fatigue state, which is beneficial to muscle recovery and regeneration and for increasing muscle size and strength [6,7]. Therefore, how to accurately identify the initial fatigue state and extreme fatigue state of the relevant muscles during exercise has become a key question in the scientific research of resistance training.

Existing research on static muscle fatigue identification is relatively mature. Generally, sensors are used to collect sEMG signals. Time and frequency domain features are extracted, and classification algorithms such as SVM\LDA\CNN\LSTM were constructed to identify muscle fatigue status based on the subjective and objective fatigue level of the human body [8,9]. According to the study, the sEMG signal of static muscle contraction has a good regularity [10,11,12]. However, during dynamic muscle contraction, because of the influence of various factors, the nonlinearity and non-stationarity of the sEMG signal increase significantly. There are various uncertainties in the time and frequency domain features, which leads to greatly reduced effectiveness in identifying muscle fatigue states [9,13]. For instance, the effectiveness of an integral electromyogram (iEMG) in characterizing muscle fatigue changes when different subjects exercise at different intensities [14]. During walking and running, mean power frequency (MPF) has not changed significantly due to the joint effect of increased temperature in muscles and increased fatigue [15]. In addition, MPF has not showed a downward trend when the trainers are engaged in medium- and low-intensity sports [16]. Therefore, the recognition rate will be affected when using time and frequency domain features to identify dynamic muscle fatigue.

Due to the complexity of physiological systems, sEMG signals have the characteristics of chaotic signals [17,18,19]. Therefore, researchers have proposed to extract the nonlinear dynamic fatigue characteristics from sEMG signals [20]. For example, multi-scale entropy [21], fractal dimension [22,23], the Lyapunov index [24], wavelet packet entropy [25,26], power spectrum entropy [27,28], etc., have been proven to be good indicators of muscle fatigue. Although many nonlinear dynamic features are more effective in identifying the fatigue state of muscles in resistance training than the time- and frequency-domain linear features, extraction of nonlinear features is a manual process. There is information loss, and the actual recognition accuracy is not high because of the amount of calculation needed for feature extraction and fatigue state classification. There is room for further improvement. For example, Karthick et al. used a support vector machine (SVM) to combine various features, feature classifiers, and feature selection techniques to obtain a variety of muscle fatigue recognition models. The highest recognition accuracy was 91%, but the recognition accuracy of most models was only 60–88% [29].

The existing research mainly focused on the identification and classification of fatigue and non-fatigue states. However, in actual training, the initial fatigue state of the trainers’ related muscles is different, and different initial muscle states need to be evaluated to match more scientific training guidance. This paper aimed to study the identification methods of different initial fatigue states and extreme fatigue states, followed by the division into four fatigue states: relaxed, a little tired, very tired, and extremely tired [30]. Knee flexion and extension training is one of the most effective methods of leg strength training. This paper constructed an isokinetic knee joint training experiment to explore an appropriate method which could recognize four muscle fatigue states in actual anaerobic training.

Therefore, works were carried out as follows to attain the above research purpose. Firstly, to avoid the impact of uncertainties in the time and frequency domain features, wavelet packet entropy and power spectrum entropy were extracted from the relevant muscle sEMG signals to verify the feasibility of characterizing different muscle fatigue. Then, two algorithms of Multi SVM and Multi LDA were constructed to classify and identify fatigue, respectively. Secondly, to avoid the loss of information in the process of feature extraction caused by the separation of feature extraction and decoding modules in traditional methods, this paper also attempted to build a CNN model based on deep learning theory, which has less data training, shorter training time and lower application cost in the later stage compared with other neural network algorithms mentioned in existing researches. Directly oriented to the original signal, CNN can extract wider, deeper, and more discriminative feature information than traditional manual feature extraction methods, so as to classify the sEMG of the target muscle according to the degree of fatigue in the isokinetic strength training of the knee joint. Finally, using experimental data, we compared the CNN exercise fatigue model with the two methods of manual fatigue feature extraction and classification (i.e., Multi-SVM and Multi-LDA) to obtain the best fatigue recognition model with higher accuracy.

## 2. Experiment Design

### 2.1. Subjects

In this experiment, 64 healthy men (*n* = 64; age: 25.8 ± 1.85 years; Height: 174.72 ± 3.88 cm; Weight: 64.75 ± 3.42 kg) with similar body sizes were recruited as the participants. Due to the influence of age and gender on local RPE [31], and considering the safety of this experiment, men of a similar age were selected as subjects. All participants were informed of the purpose of the experiment, the experimental procedure, the expected duration of each experimental stage, and the possible risks. They all signed an informed consent form.

### 2.2. Data Acquisition

In this experiment, subjects were required to perform isokinetic knee flexion and extension strength training on isokinetic testing training equipment. As the calf muscles are mainly involved in knee flexion and extension, considering the different functions of the muscles in the knee flexion and extension exercise [32,33,34], six muscles of the right leg were selected. The name, electrode location, and sEMG signal acquisition channels of the target muscles are listed in Table 1.

sEMG signals of the target muscles were collected through the MP150 multi-channel physiological signal acquisition system throughout the experiment. The German IsoMed2000 isokinetic test trainer was used as the isokinetic test trainer. The equipment is shown in Figure 1.

During the experiment, the experimenter recorded the subjective muscle fatigue through the corresponding rating of perceived exertion (RPE) scale, as listed in Table 2. In order to well reduce the cognitive difference of subjective fatigue among different subjects, the real-time fatigue of the subjects was obtained by the grade of 6–20 in this experiment, which was finally divided into 4 muscle fatigue classification labels: relaxed (6–12), a little tired (13–16), very tired (17–18), and extremely tired (19–20).

### 2.3. Experimental Process

The experimental process is shown in Figure 2.

#### 2.3.1. Pre-Experiment Preparation

To reduce the impedance, it is vital to remove the body hair of the tested parts and wipe it with alcohol to remove the surface stains. After the alcohol was air-dried, the Ag-AgCl electrodes were pasted on and connected to the sEMG acquisition channels through a wireless connection. Subsequently, the subjects randomly applied force, and the experimenter observed the collection of EMG signals and verified that the connection was feasible.

Afterwards, the subjects were instructed to sit on the isokinetic training equipment and adjust the knee joint training devices. The experimenter had made them be familiar with the usage of the equipment. At the same time, the experimenter observed the samplings of muscle strength information to ensure the normal operation of isokinetic equipment.

#### 2.3.2. Acquisition of Experimental Basic Data

According to fitness guidance, 60°/s speed can be used to train muscle endurance in knee joint strength training, and 180°/s speed can be used to train muscle explosive force [35]. Therefore, this study chose these two modes for training experiments.

First, the participants performed right leg (single leg) knee joint flexion and extension strength training at a fixed slow speed (60°/s). The participants were required to exert maximum muscle strength throughout the training, and the range of motion of the joint was set to 15°–105° (0° is knee extension to the horizontal level). A total of four continuous movements were tested without interruption (completing knee flexion and extension was considered one movement). After all the slow speed tests were completed, they rested for 10 min and then performed knee isokinetic training at a fast speed (180°/s). Similarly, a total of four continuous movements were tested without interruption [4,36,37].

In the above test of 2 groups with different speeds, the MP150 multi-channel physiological signal acquisition system was used to collect the sEMG signals synchronously and obtain the dynamic changes in sEMG during maximum voluntary contractions (MVCs) for each muscle at each speed. To eliminate the impact of individual differences, four segments of MVC data from each group were used to generate mean dynamic MVCs for each muscle.

#### 2.3.3. Knee Joint Isokinetic Training Fatigue Experiment

When training at the two speeds, the participants exerted maximum muscle strength throughout the whole 40 movements [37]. The sEMG signals of the target muscles were collected by the MP150 multi-channel physiological signal acquisition system. Meanwhile, Acknowledge4.2 was used to record data. The IsoMed 2000 isokinetic test trainer was used to record the participants’ exertion in real-time, and the experimenter recorded the participant’s perceived muscle fatigue every other movement cycle (slow for 3 s and fast for 1 s) through the RPE scale.

#### 2.3.4. Experimental Data Processing and Analysis

The experimental data were exported and analyzed by SPSS 26.0. MATLAB R2022b was used for programming recognition models.

### 2.4. Experimental Data Processing and Analysis

#### 2.4.1. Preprocessing of Experimental Data

In the above exercise fatigue experiment, the sEMG signals of the six target muscles of each participant were obtained at two exercise speeds. At the speed of 60°/s and 180°/s, the participants completed a knee flexion and extension exercise in 3 s and 1 s, respectively. After segmenting the raw sEMG data according to the starting point of each action, we performed time normalization processing. Each participant had two groups of sEMG signal sequences, one for 3 s per segment and the other for 1 s per segment. After that, the sEMG signal sequence segments at two speeds were sampled, respectively with the sampling time of 1 s and 250 ms.

Because of the large differences in the sEMG eigenvalues among the participants, the sEMG values were not suitable to be used directly for subsequent feature analysis and model input. In this study, MVC ($sEMG_{MVC}$) was used as the benchmark to preprocess the sEMG data. The calculation formula is as follows:(1)$$sEMG{\%}_{MVC}=\frac{sEMG}{sEMG_{MVC}}\times 100\%$$

After the calculation of the above formula, the preprocessed data was obtained. According to the four types of fatigue state in the experiment, the fatigue state represented by each sEMG signal at each sampling time point was calibrated.

#### 2.4.2. sEMG Signal Processing and Analysis

Based on existing research [38], this experiment extracted the PSE and WPE from the sEMG signal data and analyzed the correlation between these features and dynamic muscle fatigue. According to the number of action cycles, the sEMG signals were divided into 40 groups of action cycle data and normalized according to the $sEMG_{MVC}$. Power spectrum analysis and wavelet packet transformation were then performed on each group of sEMG signals to solve the PSE and WPE in order to assess muscle fatigue.

Both PSE and WPE were expanded based on frequency band decomposition and energy decomposition. First, the entire signal sample was decomposed into z sub-bands. The frequency band bandwidth was unequally divided, and the design was adjusted by the experimenter. The power spectrum energy $E_{x}$ (*x* = 1, 2, 3, … *z*) of each sub-band was calculated by fast Fourier transform to obtain the total power spectrum energy:(2)$$E=\overset{z}{\underset{x=1}{\sum }}E_{x}$$

Second, the probability density distribution was used to represent the energy distribution of the sEMG signal in each sub-band. The energy probability density distribution in each frequency sub-band is the ratio of the energy of each subpower spectrum to the total power spectrum energy:(3)$$P_{x}=\frac{E_{x}}{E}$$
where $P_{x}$ is the signal energy of the x-th sub-band; 0 ≤ $P_{x}$ ≤ 1 (*x* = 1, 2, 3, … *N*); and $\overset{z}{\underset{x=1}{\sum }}P_{x}=1$.

(1) PSE measures the complexity of the signal by analyzing the energy distribution of different frequency bands of the signal. According to the Shannon information entropy definition, the PSE is
(4)$$\mathrm{PSE}=-\overset{z}{\underset{x=1}{\sum }}P_{x}Iz\;P_{x}$$

A larger PSE value indicates a more uniform energy distribution of the sub-band and a more complex sEMG signal distribution. The PSE can evaluate the complexity of the sEMG and then evaluate muscle fatigue.

Take VM for example, the PSE results are shown in Figure 3. At the two speeds, the PSE showed a steady downward trend with the movement process. The PSE had a very significant negative correlation with the number of movements (*p* < 0.01), and the correlation coefficient was *r* = 0.869 (slow) and *r* = 0.818 (fast). When moving at 60°/s, the power spectrum entropy decreased slowly. Through linear fitting, the slope of the fitting curve was −0.0156. When moving at 180°/s, the slope of the linear fitting curve was −0.0204.

(2) WPE is based on wavelet transform. First, the sub-bands without subdivision are further refined and decomposed by wavelet packet analysis, and the original signal is reconstructed. Then, the energy of the reconstructed signal of each sub-band is calculated to improve the resolution of the frequency band. We first divided the signal into *z* sub-bands, denoted the x-th sub-band as $S_{x}$, and decomposed it into *y* frequency bands through the wavelet packet. Then, we have
(5)$$S_{x}^{y,\;m\;},\;m=1,\;2,\;3,\;\ldots ,\;y$$

The sub-band $S_{x}$ is
(6)$$S_{x}=A_{m=1}^{y}S_{x}^{y,\;m}$$

Then, the signal of the sub-band $S_{x}$ is reconstructed:(7)$$S_{x}\left(\tau \right)=\underset{y}{\sum }B_{y}^{N,\;x}\left\{\varphi_{x,y}\left(\tau \right)\right\}$$
where $\left\{\varphi_{x,y}\left(\tau \right)\right\}$ is the wavelet function. According to the orthogonal wavelet transform, the signal energy of each sub-band after reconstruction is obtained as
(8)$$E_{x}=\underset{y}{\sum }{\left|B_{y}^{z,\;x}\right|}^{2}$$

Similarly, according to the Shannon information entropy definition, the WPE is
(9)$$WPE=-\overset{z}{\underset{x=1}{\sum }}P_{x}IzP_{x}$$

Take VM, for example, the WPE results are shown in Figure 4. At the two speeds, the WPE also showed a steady downward trend with the movement process. The WPE had a very significant negative correlation with the number of movements, and the correlation coefficient was r = 0.853 (slow) and r = 0.829 (fast). When moving at 60°/s, through linear fitting, the slope of the fitting curve was −0.0144. When moving at 180°/s, the slope of the linear fitting curve was −0.0142.

It can be seen from the above results that, when doing isokinetic knee flexion and extension at two speeds, the PSE and WPE of VM showed a good regular decline with the deepening of fatigue, which had a good characterization of dynamic muscle fatigue, and can be used as the sEMG features to identify muscle fatigue during dynamic muscle contraction.

The data of the other five muscles were processed with the same method, and the PSE and the WPE of the five muscles’ sEMG all showed a significant negative correlation. The correlation coefficient is listed in Table 3. Therefore, the PSE and WPE of these six muscles can be used as the signal features of muscle fatigue identification in the isokinetic knee joint flexion and extension exercise. In the subsequent dynamic fatigue recognition, the PSE and WPE of these six muscles will be used as the fatigue features to build a fatigue recognition model.

#### 2.4.3. Dynamic Fatigue Recognition Based on CNN Fatigue Feature Extraction

##### Construction of the CNN Model Based on Deep Learning

sEMG signal is a mixed signal formed by the spatiotemporal superposition of the motor unit action potential at the electrode generated by the excitation of motoneurons [39]. Based on the spatiotemporal characteristics of the sEMG, this study designed a CNN structure, as shown in Figure 5. In the first convolution layer, a vector-type convolution kernel instead of a matrix-type convolution kernel was used. The single-layer convolution operation only extracts the spatial features and then the temporal features are extracted in the second convolution layer. Specifically, as shown in Figure 5, in the feature extraction, it is necessary to take into account both the temporal and spatial features of sEMG signals, and the classification part is similar to the back-propagation (BP) neural network.

Taking the 60°/s exercise as an example, the CNN design included five layers. The first layer is the data input layer; the second and third layers are convolutional layers, which complete feature extraction; the fourth and fifth layers are the fully connected layers, which, together with the eigenvalues output by the third layer, perform the classification. The specific contents of each layer of the CNN are as follows:

The first layer (I1) is the input layer. Each original input sample is a 6 × 120 matrix, where 6 represents the six target muscle channels, and 120 represents the sampling time point of each channel signal.

The second layer (C2) is the convolutional layer. This layer is locally connected to the input layer and performs spatial filtering on the original input samples. The convolution kernel size of layers I1 to C2 is 1 × 6. Ten types of filters are selected; the original input samples are convolved by each filter so there are 10 different types of feature mapping. Therefore, 10 feature maps with the size of 1 × 120 are generated. To separate the mixed spatiotemporal information, the convolution kernel is set as a vector instead of a matrix so that only spatial features are included in the features after the convolution operation.

The third layer (C3) is the convolution-pooling layer. This layer realizes the temporal feature extraction of the sEMG signal by adding local links and weight sharing. To prevent overfitting, we set the convolution kernel size to 1 × 10 and the convolution stride to 10 to reduce the number of parameters and implement pooling operations. For the 10 feature maps in the C2 layer, four convolution filters are used for each. After the mapping, the C3 layer generates 40 different feature maps, each with a size of 1 × 12.

The fourth layer (F4) is the fully connected layer. This layer fully connects the C3 layer and the O5 layer to generate the classification. In this design, the number of neurons is set to 100.

The fifth layer (O5) is the output layer. This layer contains four neurons representing the four types of fatigue: no fatigue (RPE = 6–12, relaxed), mild fatigue (RPE = 13–16, a little tired), severe fatigue (RPE = 17–19, very tired), extremely tired (RPE = 20, extremely tired).

For the 180°/s sample, according to the data preprocessing, the sampling time was 250 ms, and the parameters of each layer of the CNN model were modified accordingly. Each original input sample of the I1 layer is a 6 × 160 matrix. The convolution kernels from the I1 layer to the C2 layer are unchanged, and the size of the feature map of the C2 layer is correspondingly adjusted to 1 × 160. The size of the feature map of the C3 layer is 1 × 16. Other settings remain consistent with the CNN model for 60°/s.

Using the above method, the CNN dynamic fatigue model for the two speeds was established.

##### Learning and Training of the CNN Model

In this study, the BP method was used to complete the learning and training of the above-mentioned CNN. First, we input the preprocessed training data and obtained the activation value of each neuron according to the forward calculation method. We then carried out the reverse error calculation. Each weight and bias gradient were obtained according to the error. Finally, the original weight and bias values were adjusted based on the new weight and bias gradient.

We set β(*l*, *p*, *q*) as the expression of any neuron in the CNN, where *l* represents the layer number; *p* represents the *p*th feature map of that layer; and *q* represents the *q*th neuron in that feature map. Based on this, $x_{p}^{l}\left(q\right)$ and $y_{p}^{l}\left(q\right)$ represent the input and output data of a neuron, respectively. We constructed an activation function as follows:(10)$$y_{p}^{l}\left(q\right)=f\left[x_{p}^{l}\left(q\right)\right]$$
where f(x) is the activation constant. The activation function of the C2 and C3 layers of the CNN network was set to be a hyperbolic tangent function as follows:(11)f(x)=atanh(bx)
where a and b are constants. We took a = 1.71159 and b = 2/3 [40]. The activation function of the F4 layer and the O5 layer of the CNN network was a sigmoid function:(12)$$f\left(x\right)=\frac{1}{1+exp^{-x}}$$

In the CNN, data transmission was performed between neurons in each layer. The specific transmission relation was as follows:

The first layer (I1), with 6 channels × 120 sampling time points, is represented as $Y_{j,t}$, where *j* represents the channel number, and *t* represents the sampling time point.

In the second layer (C2), the feature map of the I1 layer is convolved by a convolution kernel with a pre-defined size of 6 × 1 activated by the hyperbolic tangent function. The feature map of the output C2 layer can be obtained, and the transfer function was
(13)$$y_{p}^{2}\left(q\right)=f\left[\overset{j<6}{\underset{j=1}{\sum }}Y_{j,\;q}\times k_{p}^{2}+b_{p}^{2}\left(q\right)\right]$$
where $k_{p}^{2}$ is the convolution kernel of [6 × 1], and $b_{p}^{2}\left(q\right)$ is the bias.

The data transfer method of the third layer (C3) was similar to the C2 layer:(14)$$y_{p}^{3}\left(q\right)=f\left\{\overset{j\leq 10}{\underset{j=1}{\sum }}y_{p}^{2}\left[\left(q-1\right)\times 10+j\right]\times k_{p}^{3}+b_{p}^{3}\left(q\right)\right\}$$
where $k_{p}^{3}$ is the convolution kernel of [10 × 1], and $b_{p}^{2}\left(q\right)$ is the bias.

In the fourth layer (F4), all the neurons are fully connected with all the neurons in the C3 layer. The connection function was as follows:(15)$$y^{4}\left(q\right)=f\left[\overset{j\leq 40}{\underset{j=1}{\sum }}\overset{w\leq 6}{\underset{w=1}{\sum }}y_{j}^{3}\left(w\right)\omega_{j}^{4}\left(w\right)+b^{4}\left(q\right)\right]$$
where $\omega_{j}^{4}\left(w\right)$ is the weight of the connections between the C3 layer neurons and the F4 layer neurons, and $b^{4}\left(q\right)$ is the bias.

In the fifth layer (O5), all the neurons are fully connected with all the neurons in the F4 layer. The connection function is
(16)$$y^{5}\left(q\right)=f\left[\overset{j\leq 100}{\underset{j=1}{\sum }}y^{4}\left(j\right)\omega^{5}\left(j\right)+b^{5}\left(q\right)\right]$$
where $\omega_{j}^{5}\left(w\right)$ is the weight of the connections between the F4 layer neurons and the O5 layer neurons, and $b^{4}\left(q\right)$ is the bias.

Next, we initialized the CNN weights and biases to provide preconditions for the effective training and convergence of the CNN network. First, the connection weights and biases in the initialized CNN were evenly distributed in the interval $\left[\pm \frac{1}{n{\left(l,p,q\right)}_{Ninput}}\right]$, where $n{\left(l,p,q\right)}_{Ninput}$ represents the number of neurons in the previous layer of the *l*th layer that are connected to the *q*th neuron in the *p*th feature map of the *l*th layer. We set the learning rate of the C2 and C3 layers to γ [41], and it was calculated as follows:(17)$$\gamma =\frac{2\lambda }{Nshared_{p}^{l}\sqrt{n{\left(l,p,q\right)}_{Ninput}}}$$
where $Nshared_{p}^{l}$ represents the number of shared-weight neurons in the *p*th feature map of the *l*th layer in the network. Finally, the formula for setting the learning rate γ between F4 and O5 was as follows:(18)$$\gamma =\frac{\lambda }{\sqrt{n{\left(l,p,q\right)}_{Ninput}}}$$

Thus, the weights and biases of the connections between the CNN layers were set. In order to minimize the fatigue recognition error, the gradient descent method was used to adjust the weights and biases. We set the maximum number of network iterations to 10,000. In the CNN training process, by analyzing the loss function, we determined whether the network had converged, and finally, the optimal fatigue recognition model was selected.

#### 2.4.4. Experimental Sample Construction

There were 64 subjects in the experiment. The data of 60 random subjects were selected from 64 subjects. The experimental data of the 60 subjects were divided into 5 segments according to the ratio of 3:1:1. Three segments (60%) of the data were used as training data; one segment (20%) of the data was used as the validation data; one segment (20%) of the test data was used as the test data. The training data was used for constructing the model, while the validation data was used to select the optimal parameters of the model, and the test data was used to evaluate the model recognition rate. The data of the rest 4 subjects was used for further model validation and was not utilized in the training.

In this study, while building the CNN model to identify muscle fatigue status, Multi-SVM and Multi-LDA were constructed based on PSE and WPE to identify fatigue. Three types of models were trained with the same training data and were tested on the same test data.

(1) Multi-SVM: Used PSE and WPE of 6 muscles’ sEMG as features, and then classified them by SVM classifier using the Gaussian kernel function. The kernel function formula is as follows.
(19)$$K\left(x,\;y\right)=exp\left(\frac{ǀǀx-yǀǀ}{2\sigma^{2}}\right)$$

(2) Multi-LDA: The sEMG’s PSE and WPE of 6 muscles were also extracted, and then classified by the Multi-LDA classifier. Mark the characteristics of the input sEMG signal as *x_i_* (*i* = 1, 2, 3, …., *n*), set the input sample set as $X=\left\{\left({𝓍}_{1},{𝓎}_{1}\right),\ldots ,\left({𝓍}_{n},{𝓎}_{n}\right)\right\}$, where ${𝓎}_{n}$ corresponds to the LDA classification label as *X_a_* (*a* = 1, 2, …, 4). Based on the above marks, the four-classification algorithm formula of Multi-LDA’s is as follows:(20)$$\mathrm{LDA}\;(X_{1},X_{2},X_{3},X_{4})=\frac{w^{T}S_{B}\left(X_{1},X_{2},X_{3},X_{4}\right)w}{w^{T}S_{W}\left(X_{1},X_{2},X_{3},X_{4}\right)w}$$

In the formula, *S_B_* represents the between-class scatter matrix, *S_W_* represents the within-class scatter matrix. The Multi-LDA is maximized to achieve the maximum *S_B_* and the minimum *S_W_*, so that the dimension-reduced classification sample obtains the maximum inter-class distance and the minimum intra-class distance.

#### 2.4.5. Evaluation Index of Exercise Fatigue Identification

In this study, accuracy (Acc) and the receiver operating characteristic (ROC) were used for evaluation.

Accuracy is the ratio of the number of samples truly classified by the model to the total number of samples participating in the classification, which can be expressed as:(21)$$\mathrm{Acc}=\frac{TP+TN}{TP+TN+FP+FN}$$
where *TP* is true positives, *TN* is true negatives, *FP* is false positives, *FN* is false negatives.

This experiment is a multi-classification problem. Thus, we used micro-Precision ($Pre_{m}$), micro-Recall ($Rec_{m}$), and micro-F1score ($F1_{m}$) to assess the identification of fatigue states by the three classification models. In the multi-classification case, the value of the three is equal and the same as accuracy. At this point, accuracy is the ratio of the sum on the diagonal to the total number of samples, so it can be understood as:(22)$$\mathrm{Acc}=\frac{\sum_{i=1}^{n}TP_{i}}{All}=Pre_{m\;}=Rec_{m}=F1_{m}$$
where $TP_{i}$ is the number of truly classified samples in each category, and *All* is the total number of samples participating in the classification.

The closer the ROC curve is to the upper-left boundary, the better the performance of the classification model.

## 3. Results and Discussion

### 3.1. CNN Training Process and Results

The training data were used to train the CNN model. A convergent network model was obtained after training. Taking 60°/s as an example, the loss curve in model training is shown in Figure 6. The abscissa is the number of iterations, and the ordinate is the loss value. The dotted and solid lines indicate the percentages of the loss value of the CNN network at different iterations when the model is trained with the training data and validation data, respectively.

As shown in Figure 6, after 3,093 iterations, the validation loss reached the lowest point at 1.8297% and then slightly rose and remained stable with an increasing number of iterations, whereas the training loss decreased slightly and remained stable with an increasing number of iterations. It can be seen that the CNN fatigue recognition model after the 3,093rd iteration is the optimal fatigue classification model for the 60°/s exercise. The 180°/s exercise’s optimal fatigue recognition model can be determined in the same way.

### 3.2. Exercise Fatigue Recognition Results Based on Test Samples

To evaluate the performance of the proposed fatigue recognition model, three models were applied to the test samples: (1) Multi-SVM; (2) Multi-LDA; (3) CNN.

The confusion matrix of subject classification of the test data under three methods is shown in Figure 7. The numbers in the matrix diagonal grid (gray grid) represent the average percentage of the number of correctly classified samples for all subjects; the numbers in the off-diagonal grid (white grid) represent the average percentage of the number of incorrectly classified samples for all subjects.

Table 4 lists the identification of the 3 models testing at test samples. For the samples of 60°/s, the overall recognition accuracy of the CNN model was 91.38%, which was higher than that of Multi-SVM (90.17%) and Multi-LDA (88.85%). For the samples of 180°/s, the overall recognition accuracy of the CNN model was 89.87%, which was higher than which of Multi-SVM (89.21%) and Multi-LDA (87.69%).

Figure 8 and Figure 9 show the fatigue recognition performance of the three models in the samples of the two speeds. It can be intuitively learned from the figures that the area under the ROC curve of the CNN model is larger than that of the Multi-SVM and Multi-LDA models, i.e., the CNN model had better performance than the other two models.

To evaluate the interaction between the classification models and fatigue categories and their influences on the classification results, analysis of variance (ANOVA) was used in this paper to conduct variance analysis on three classification models × 4 fatigue categories, and the confidence level was set to 95%. The results of the ANOVA showed that there was no interaction between the classification model and the fatigue category (*p* > 0.05). The classification model had a significant effect on the classification results (F = 6.32, *p* < 0.01), whereas the exercise fatigue category had no significant effect on the classification results (*p* > 0.05).

Because CNN can directly process the original signal, it can extract more useful features and at the same time reduce the loss of information during data processing; so, it shows a higher recognition accuracy than the other two classification models [42].

### 3.3. Verification of the CNN Exercise Fatigue Recognition Model Based on New Samples

According to the comprehensive evaluation of the above test results, the fatigue recognition model based on CNN exercise fatigue feature extraction had better recognition performance than other models. To further verify the practical application of this model, the rest four participants’ experimental data were input into the CNN fatigue recognition model. The results showed that the average recognition accuracy of the CNN model for the four participants was 86.75% ± 4.01% for 60°/s movement and 84.86% ± 4.17% for 180°/s movement. The ROC mean curves of the classification results at the two speeds are shown in Figure 10.

Compared with the former recognition results of the test data, the CNN model’s recognition accuracy for the rest of the four participants had descended. The reason may be that the training sample was not large enough. After adding the data of the four participants to the training, the model was tested again. We found that the recognition accuracy increased to 92.16%. Therefore, before future application, the training samples should be increased to obtain a high recognition accuracy.

## 4. Conclusions

In this paper, we constructed three types of fatigue recognition models, the first two were Multi-SVM and Multi-LDA based on manual feature extraction, and the third was the CNN recognition model using the original sEMG signal. In this experiment, two nonlinear kinetic indices of PSE and WPE were manually extracted from sEMG and used to train Multi-SVM and Multi-LDA. Simultaneously, the sEMG data was directly used to learn and train CNN models, and then, the recognition accuracy of the three models was compared. The results showed that for the test data, the average recognition accuracy (91.38%) and the area under the ROC curve of the CNN fatigue recognition model were larger than those of the other two recognition models. The results indicate that the CNN model has better classification performance. At last, after training with four more samples, the recognition accuracy of the CNN recognition model reached 92.16%. The results verified that the CNN dynamic fatigue state recognition model based on subjective and objective information feedback presented satisfactory recognition performance.

This paper makes several contributions to the literature. First, this study realizes the recognition of four muscle fatigue states in resistance strength training. No other study has been able to do this in the past. This will lay a foundation for the design and development of intelligent fitness equipment and personalized fitness guidance. Second, the results of this study help provide two nonlinear kinetic indices of PSE and WPE, which could be used in further study of the higher rate of accurate classifiers. Third, our study gives a dynamic muscle fatigue recognition method with relatively higher accuracy and lower time cost, which could be better commercialized.

## 5. Future Prospects

This research can provide theoretical support for the design and research of wearable devices in the future and help to develop a more effective fitness training model. However, this experiment has several limitations: (1) Only knee flexion and extension exercises were used as the target, and other training movements need to be studied; (2) Isokinetic training was investigated in this experiment, so other training methods in anaerobic training need to be studied; (3) The number of participants was small. Subsequent research should increase the number of participants to train the recognition model; (4) The classifiers of this research were limited. In addition to the classifiers used in this study, there are ANN, LSTM, GSVCM, etc. The practical application effect of these classifiers remains to be verified. In the future, more research about these classifiers will be carried out.

## Figures and Tables

**Figure 1 healthcare-10-02292-f001:**
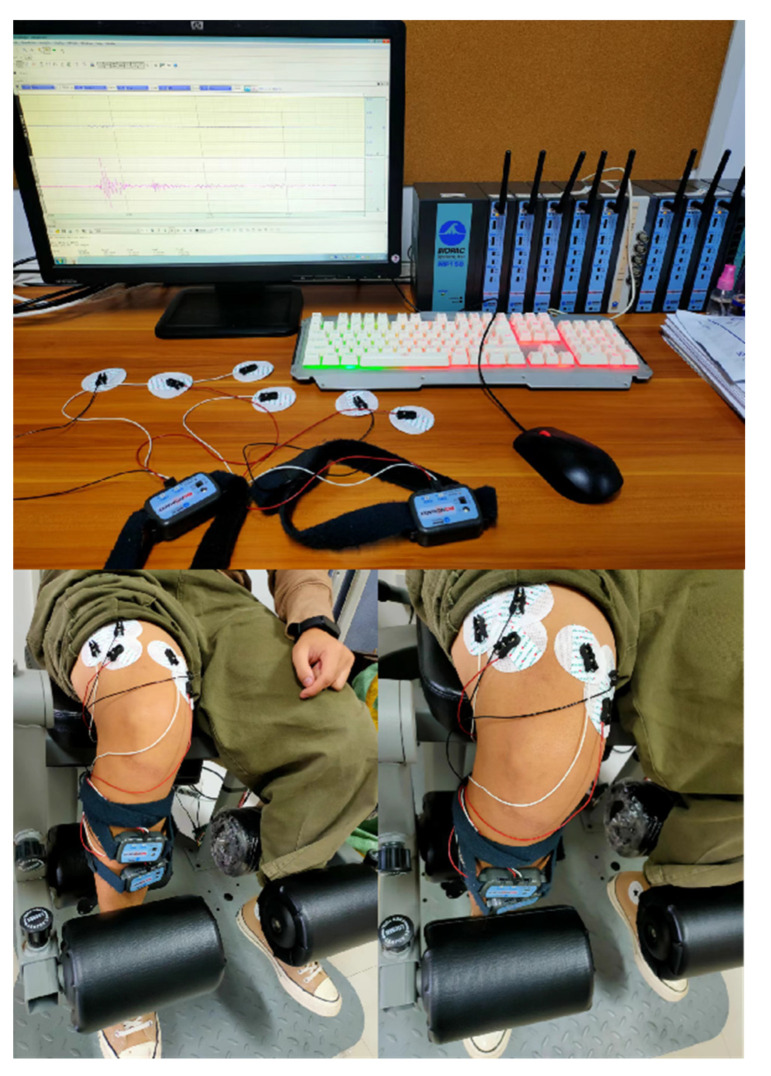
Experiment equipment, process, and scene recording.

**Figure 2 healthcare-10-02292-f002:**
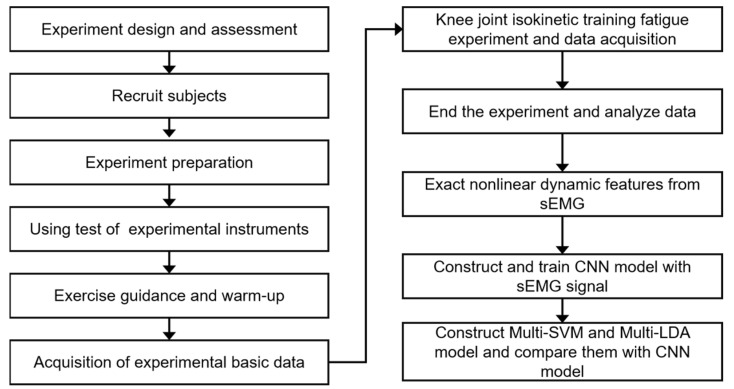
The experimental process.

**Figure 3 healthcare-10-02292-f003:**
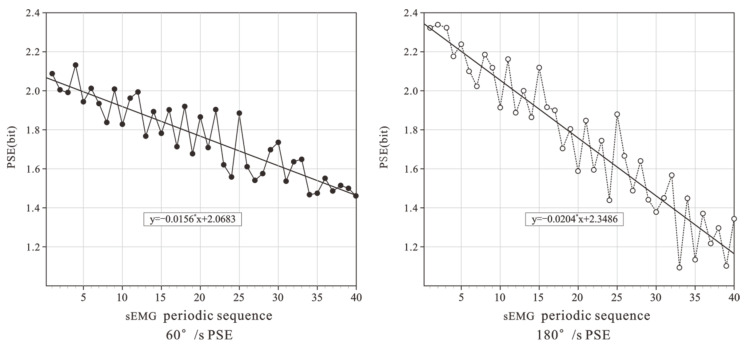
PSE of VM at 2 speeds.

**Figure 4 healthcare-10-02292-f004:**
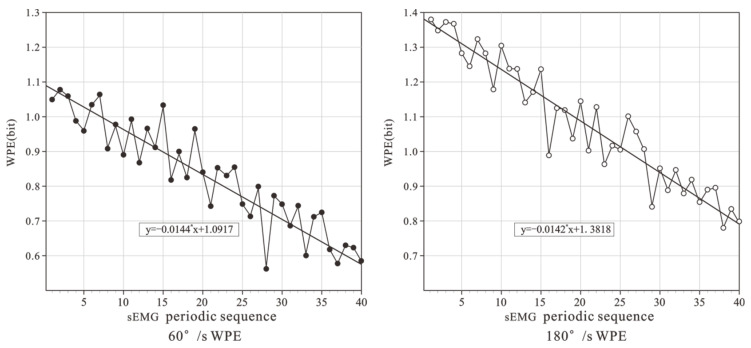
WPE of VM at 2 speeds.

**Figure 5 healthcare-10-02292-f005:**
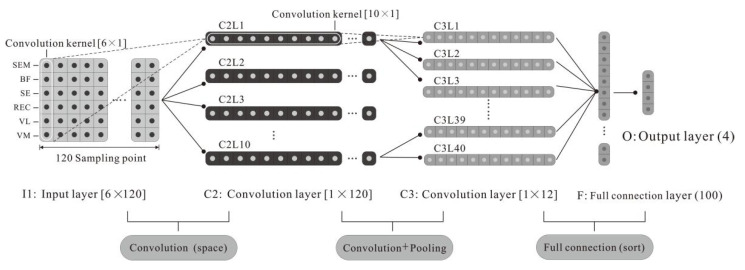
CNN structure diagram based on the classification of sEMG signals of exercise fatigue.

**Figure 6 healthcare-10-02292-f006:**
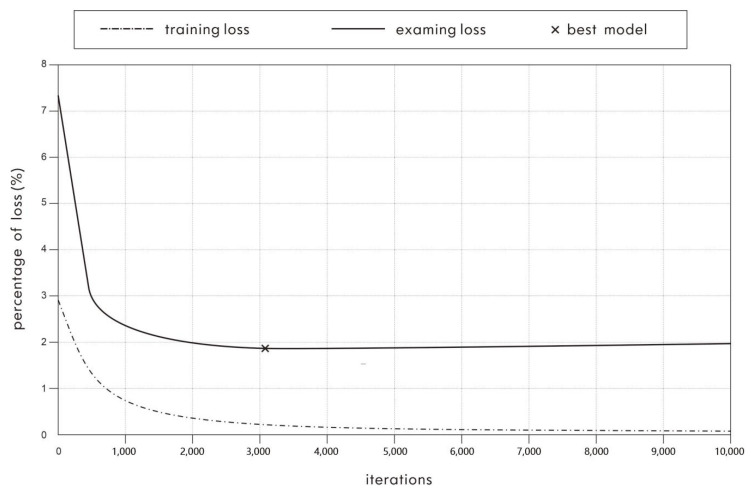
The loss curve of CNN model in 60°/s training.

**Figure 7 healthcare-10-02292-f007:**
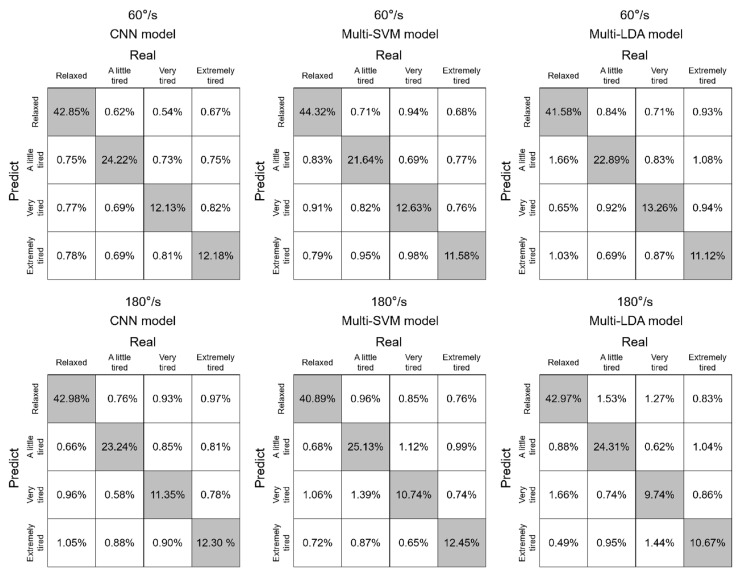
Confusion matrix results of 3 models at 2 speeds.

**Figure 8 healthcare-10-02292-f008:**
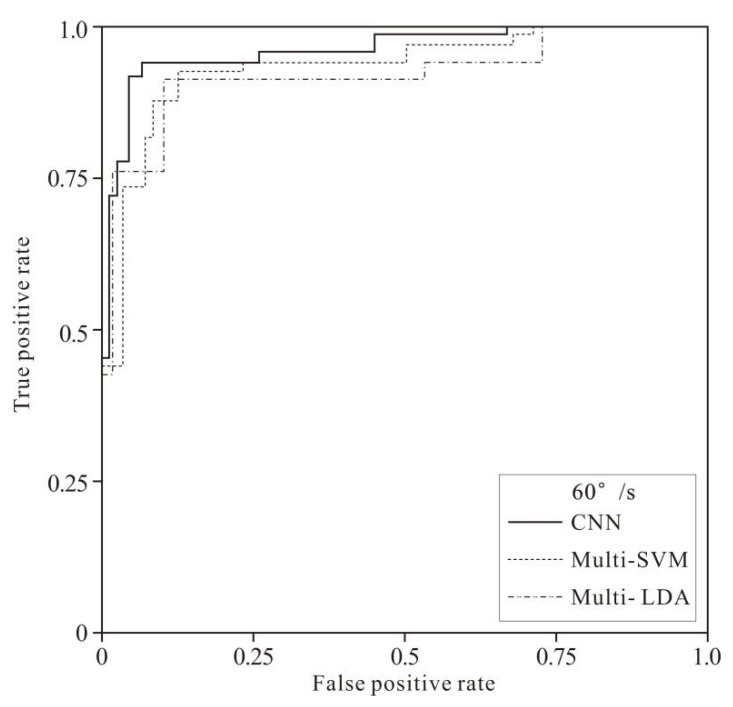
The ROC curves of 60°/s-test sample classification by 3 models.

**Figure 9 healthcare-10-02292-f009:**
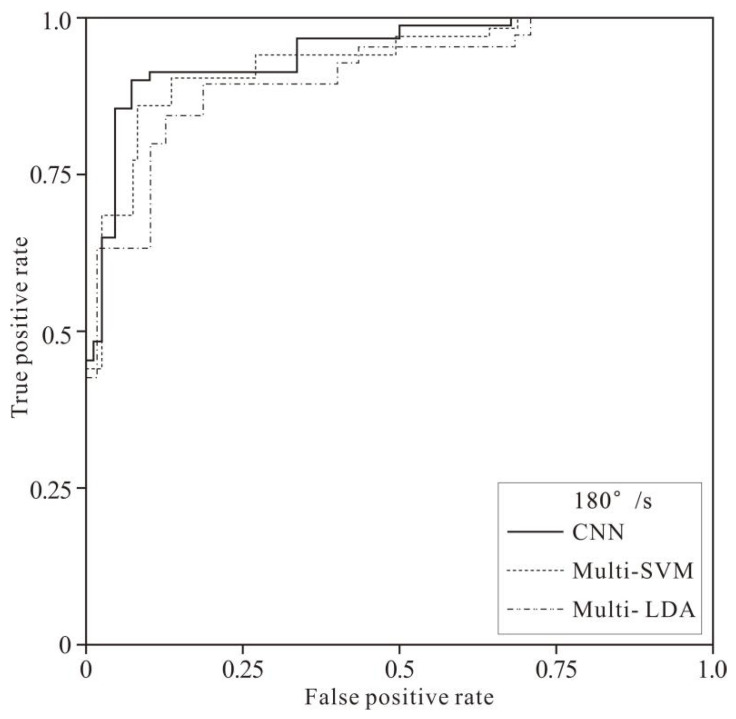
The ROC curves of 180°/s-test sample classification by 3 models.

**Figure 10 healthcare-10-02292-f010:**
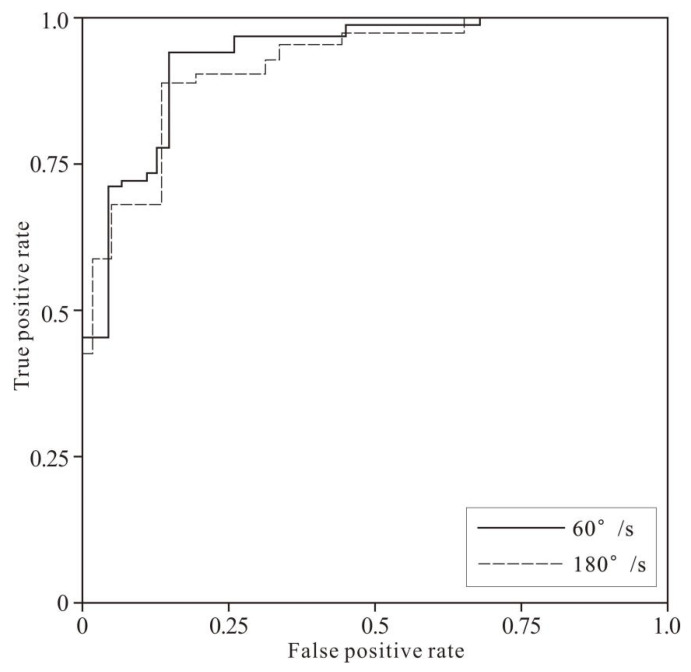
The ROC curves of 4 rest samples classification under 2 speeds by CNN.

**Table 1 healthcare-10-02292-t001:** Muscle, the electrode location, acquisition channels.

Name of the Target Muscle	Vastus Medialis (VM)	Biceps Femoris (BF)	Vastus Lateralis (VL)	Rectus (REC)	Semitendinosus (SEM)	Semimembranosus (SE)
Electrode location	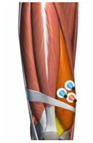	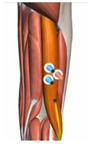	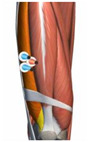	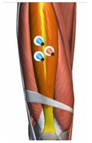	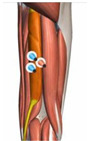	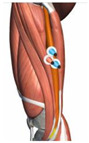
acquisition channels	sEMG-2R-1	sEMG-2R-2	sEMG-2R-3	sEMG-2R-13	sEMG-2R-14	sEMG-2R-15

**Table 2 healthcare-10-02292-t002:** Subjective RPE scale.

Evaluation Grade	Subjective Exercise Fatigue	Classification Label
6	Not hard at all	Relaxed
7	Extremely relaxed	
8		
9	Very relaxed	
10		
11	Relaxed	
12		
13	A little tired	A little tired
14		
15	Tired	
16		
17	Very tired	Very tired
18		
19	Extremely tired	Extremely tired
20	Try the best	

**Table 3 healthcare-10-02292-t003:** The Pearson correlation between the nonlinear index and the number of motion.

Speed	60°/s		180°/s	
Index	PSE	WPE	PSE	WPE
SEM	0.663 * (*p* = 0.017)	0.746 * (*p* = 0.034)	0.672 * (*p* = 0.045)	0.690 * (*p* = 0.029)
BF	0.732 ** (*p* = 0.003)	0.767 ** (*p* = 0.002)	0.698 ** (*p* = 0.009)	0.716 ** (*p* = 0.004)
SE	0.681 * (*p* = 0.031)	0.722 * (*p* = 0.026)	0.712 * (*p* = 0.022)	0.733 * (*p* = 0.014)
REC	0.904 ** (*p* = 0.002)	0.869 ** (*p* = 0.003)	0.831 ** (*p* = 0.006)	0.844 ** (*p* = 0.008)
VL	0.791 ** (*p* = 0.007)	0.812 ** (*p* = 0.004)	0.764 ** (*p* = 0.002)	0.775 ** (*p* = 0.003)
VM	0.876 * (*p* = 0.026)	0.865 * (*p* = 0.012)	0.823 ** (*p* = 0.006)	0.817 ** (*p* = 0.009)

**: At 0.01 level, the correlation is very significant; *: At 0.05 level, the correlation is significant.

**Table 4 healthcare-10-02292-t004:** The Accuracy of the fatigue identification in 60°/s and 180°/s training by 3 models.

Accuracy		
	60°/s	180°/s
CNN	91.38%	89.87%
Multi-SVM	90.17%	89.21%
Multi-LDA	88.85%	87.69%

## Data Availability

Involving confidentiality agreement, the data will not be disclosed.

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
