# Peer review of "Research on the Recognition of Various Muscle Fatigue States in Resistance Strength Training"

_healthcare, 2022, doi:10.3390/healthcare10112292_

Round 1
Reviewer 1 Report
The authors introduced a CNN recognition model to extract surface EMG features and classify muscle states of the target muscled in the isokinetic strength training of the knee joint. The reviewer has some concerns that need to be considered. These can be summarised hereunder.
1) As the authors indicated in page 2, CNN has been widely utilised in recognition-related fields. This also applies to fatigue muscle recognition. For instance, Wang et al. (2021) introduced a new muscle fatigue recognition model based on the long short-term memory (LSTM) network. The new approach was compared with CNN, SVM, and other classification models proposed by scholars. The approach outperformed the CNN introduced. Shouan et al. (2021) presented an end-to-end dynamic gesture recognition method based on CNN. Wang et al. (2020) proposed a muscle fatigue classification method based on surface electromyography (sEMG) on Improved Wavelet Threshold and CNN-SVM. These are just examples. Thus, the novelties need to be clarified with regards to the utilisation of CNN as a muscle recognition model.
Wang et al., 2021. A Muscle Fatigue Classification Model Based on LSTM and Improved Wavelet Packet Threshold. Sensors 21(19), doi: https://doi.org/10.3390/s21196369.
Shouan et al., 2021. Dynamic Hand Gesture Recognition via Electromyographic Signal Based on Convolutional Neural Network. Conference Proceedings - IEEE International Conference on Systems, Man and Cybernetics, doi: https://doi.org/10.1109/SMC52423.2021.9658997.
Wang et al. 2020. Recognition of Muscle Fatigue Status Based on Improved Wavelet Threshold and CNN-SVM. IEEE Access, doi: https://10.1109/ACCESS.2020.3038422.
2) The presented critical literature review is limited. For instance, none of the references above have been analysed by the authros. The authors should perform a more comprehensive critical literature review with regards to fatigue muscle recognition and the utilisation of data-driven methodologies. How the proposed methodology compares to other identified methods? Which are the advantages, disadvantages, and limitations of the identified methods?
3) In page 2 the authors mention there are various uncertainties in the time and frequency domain features. Various is ambiguous. Please include information about such uncertainties.
4) There is a typo in “2. Experiment Desing”. Please consider proofreading.
5) The authors stated in the experiment design section that the physical fitness of the selected participants was relatively close. Thus, how does the model generalise? For instance, only 60 healthy men were recruited as participants. What about other type of subjects, such as women or other groups of men?
6) The figures in section “2. Experiment Design” are barely described.
7) The reviewer finds confusing the table 2 and the categories utilised by the authors in the methodology. The authors stated in page 6 that the muscle fatigue state was dividied into four categories: relaxed, a little tirred, very tired, and extremely tired. Why did the authors only considered these four categories? Could the authors provide further information about the evaluation grade? For instance, did the authors subdivided the category a little tired as indicated in the evaluation grade (13 and 14 in this case). If the answer is no, why? Also, provide more information about how this labelling process was performed.
8) Please justify the selection of each parameter set in the section “2.3. Experimental Process”. For instace, why only two speeds were considered? And why these specific two (60º/s and 180º/s). Which was the motiviation to perform eight tests?
9) The authors indicated in page 5 that a total of four movements were tested continuously without interruption. Please provide more information about these movements.
10) A diagram of the acquisition of experiment data would be helpful to the reader for a better understanding of the test scenarios. Analogously, a diagram of the overall methodology would be helpful for the readers.
11) Please consider rephrasing sentence of line 141 page 5 ([...] Acknowledge4.2 [...]).
12) More than 10 releases have been made of MATLAB since the release of MATLABv2016. This also applies to SPSS 19.0. The software utilised should be up to date unless the usage of older releases is justified. Moreover, please specify the tasks implemented by these softwares.
13) In line 150 of page 5 the authors stated that “Based on existing research, [...]”. Please provide more information about the research considered by the authors to sustain such a fact. Analogously, the extraction of both PSE and WPE should be justified.
14) The authors indicated in page 5 that “this experiment extracted the PSE and WPE form the sEMG signal and analyzed the correlation betwen these features and dynamic muslce fatigue”. However, there is no evidence that the correlation mentioned was performed within the manuscript. Please provide further information about this process and the results obtained.
15) If the reviewer understood correctly, a total of six acquisition channels were considered. Each channel recorded a sEMG. Then PSE and WPE were extracted from the sEMG in order to assess muscle fatigue. How is this assessment performed? Why is this step requried? How is this related to the CNN methodology part? Section “2.4.1. sEMG signal processing and analysis” is confusing with the information currently presented in the manuscript.
16) Was there any challenge with regards to missing values and errors introduced by the experimenter when determining the muscle fatigue state and acquiring the sEMG signals? How did the authors address that?
17) A comparative study was performed to validate the performance of CNN. Specifically, the SVM and LDA models were considered. Please provide adequate justifications on the implementation of these models as part of the comparative study. Moreover, other state-of-the art models should be also considered to ensure credibility of the comparative study. For instance, how your model compares to the one proposed by Wang et al. (2021) (please see point 1 for full reference).
18) It seems that references are missing in line 222 of page 7 ([…] neurons []).
19) Please justify the selection of the architecture of the CNN and the shape of the inputs. For instance, why did the authors decide to use a unique full connection layer? Why were a total of 120 sampling points considered to define the input shape of the CNN? Did the location of the six target muscle channels in the input matrix (current definition: [SEM, BF, SE, REC, VL, VM]) influence in the accuracy of the model? The authors created a total of two CNN models: one for the 60º/s, and another for the 180º/s. Did they present the same architecture? Why?
20) The consideration of the confusion matrix as evaluation metric is recommended.
21) The reviewer considers that the results section needs to be expanded. For instance, more information about the results from the data acquisition and data pre-processing processes needs to be provided (e.g., visualisations of the signals, descriptive statistics, correlation analysis, etc.). Also, more information about the dataset needs to be provided. For instance, the number of samples per category and test. Is the dataset imbalanced? How did the authors address this issue?
22) Multi-SVM, and Multi-LDA were implemented as part of the comparative study. However, there is no evidence about the hyperparameter optimisation performed by the authors. For instance, how did the authors estimate the kernel, gamma, and C hyperparameters of multi-SVM?
23) A table providing the results for each model and evaluation metric is recommended.
24) Four additional participants were considered to further verify the practical application of the model. Please provide the subject basic informaiton of these participants.
25) The authors stated that after adding the additional participants to the training the accuracy increased to 92.16%. Please indicate if this 92.16% relates to the test set or to these four participants, as if these refers to the four participants there is a clear biased evaluation.
26) Which are the disadvantages and limitations of the proposed model?
Author Response
Sincerely , we appreciate your valuable comments. About your questions, after careful thinking, we found that our research still has some deficiencies and had made corresponding improvements. The following will be the answers to your questions.
Point 1:As the authors indicated in page 2, CNN has been widely utilised in recognition-related fields. This also applies to fatigue muscle recognition. For instance, Wang et al. (2021) introduced a new muscle fatigue recognition model based on the long short-term memory (LSTM) network. The new approach was compared with CNN, SVM, and other classification models proposed by scholars. The approach outperformed the CNN introduced. Shouan et al. (2021) presented an end-to-end dynamic gesture recognition method based on CNN. Wang et al. (2020) proposed a muscle fatigue classification method based on surface electromyography (sEMG) on Improved Wavelet Threshold and CNN-SVM. These are just examples. Thus, the novelties need to be clarified with regards to the utilisation of CNN as a muscle recognition model.
Wang et al., 2021. A Muscle Fatigue Classification Model Based on LSTM and Improved Wavelet Packet Threshold. Sensors 21(19), doi: https://doi.org/10.3390/s21196369.
Shouan et al., 2021. Dynamic Hand Gesture Recognition via Electromyographic Signal Based on Convolutional Neural Network. Conference Proceedings - IEEE International Conference on Systems, Man and Cybernetics, doi: https://doi.org/10.1109/SMC52423.2021.9658997.
Wang et al. 2020. Recognition of Muscle Fatigue Status Based on Improved Wavelet Threshold and CNN-SVM. IEEE Access, doi: https://10.1109/ACCESS.2020.3038422.
Reply :
The research in this paper came from the key technology research of major technology project in Zhejiang, China. It mainly studied muscle fatigue status identification in resistance strength training in order to lay a foundation for the follow-up design of fitness equipment with scientific fitness guidance function as well as provide data basis for professional fitness coaches to assist in fitness training. Considering the commercial feasibility, the accuracy of muscle fatigue identification was not the only goal that being ultimately pursued. Its cost was also taken into consideration. The training amount, training time, data storage space and hardware cost of the algorithm also needed to be comprehensively considered.
LSTM is a version of RNN, which can well depict sequence data with spatio-temporal correlation, and can predict what will happen next based on event sequences. CNN has a good recognition effect in image recognition, which can accurately identify and classify targets in real time. Comparatively, the CNN model has a lower training cost and a relatively short time, so this paper is prioritized to consider the CNN model. As for whether LSTM is better than CNN in identifying various muscle fatigue states of resistance strength training in real time and accurately, we will do more detailed research on it in the future.
Point 2:The presented critical literature review is limited. For instance, none of the references above have been analysed by the authors. The authors should perform a more comprehensive critical literature review with regards to fatigue muscle recognition and the utilisation of data-driven methodologies. How the proposed methodology compares to other identified methods? Which are the advantages, disadvantages, and limitations of the identified methods?
Reply :
As described in lines 42-80, existing research on static muscle fatigue identification is relatively mature. Sensors are used to collect sEMG signals. Time and frequency domain features are extracted, and a classification algorithms such as SVM\LDA\CNN\LSTM were constructed to identify muscle fatigue status based on the subjective and objective fatigue level of the human body. However, the current research mainly focused on the identification and classification of fatigue and non fatigue states. In actual training, the initial fatigue state of the trainers' related muscles is different, so different initial muscle states need to be evaluated to match more scientific training guidance. In order to determine the initial muscle fatigue state of the trainers, the fatigue degree is divided into four categories: relaxed, slightly tired, very tired, and extremely tired, to match more scientific and reasonable training guidance.
Point 3:In page 2 the authors mention there are various uncertainties in the time and frequency domain features. Various is ambiguous. Please include information about such uncertainties.
Reply :
As described in lines 51-58, the effectiveness of integral electromyogram (iEMG) in characterizing muscle fatigue changes when different subjects exercise at different intensities. During walking and running, mean power frequency (MPF) did not change significantly due to the joint effect of increased temperature in muscle and increased fatigue. In addition, MPF did not show a downward trend when the trainers were engaged in medium and low intensity sports. Therefore, the recognition rate will be affected when using time and frequency domain features to identify dynamic muscle fatigue.
Point 4:There is a typo in “2. Experiment Desing”. Please consider proofreading.
Reply :
The typo in “2. Experiment Design” has been corrected.
问题5:The authors stated in the experiment design section that the physical fitness of the selected participants was relatively close. Thus, how does the model generalise? For instance, only 60 healthy men were recruited as participants. What about other type of subjects, such as women or other groups of men?
Reply :
As described in lines 99-104, we recruited 64 men of similar age and body size as subjects. Reference [31] indicate that both gender and age factors can affect the subjective RPE. Considering the safety of this experiment, men of similar age were selected as subjects.
Point 6:The figures in section “2. Experiment Design” are barely described.
Reply :
The problem in “2. Experiment Design” has been improved after the modification.
Point 7:The reviewer finds confusing the table 2 and the categories utilised by the authors in the methodology. The authors stated in page 6 that the muscle fatigue state was divided into four categories: relaxed, a little tired, very tired, and extremely tired. Why did the authors only considered these four categories? Could the authors provide further information about the evaluation grade? For instance, did the authors subdivided the category a little tired as indicated in the evaluation grade (13 and 14 in this case). If the answer is no, why? Also, provide more information about how this labelling process was performed.
Reply :
As described in lines 119-124, based on references [30], in order to well reduce the cognitive difference of subjective fatigue among different subjects, the real-time fatigue of the subjects was obtained by the grade of 6-20 in this experiment, which was finally divided into 4 muscle fatigue classification labels: relaxed (6-12), a little tired (13-16), very tired (17-18), and extremely tired (19-20).
Point 8:Please justify the selection of each parameter set in the section “2.3. Experimental Process”. For instance, why only two speeds were considered? And why these specific two (60º/s and 180º/s). Which was the motivation to perform eight tests?
Reply :
As described in lines 145-147, According to fitness guidance, 60 °/s speed can be used to train muscle endurance in knee joint strength training, and 180 °/s speed can be used to train muscle explosive force. Therefore, this study chose these two modes for training experiments.
Eight tests were corrected to 2 group of tests with different speeds. Every group tested 4 continuously movements in order to obtain mean dynamic MVCs for each muscle and eliminate the impact of individual differences.
Point 9: The authors indicated in page 5 that a total of four movements were tested continuously without interruption. Please provide more information about these movements.
Reply :
Four continuously movements are four movements for each group as described in question 8. At the set speed, the subjects completed four continuously knee flexion and extension movements without interruption.
Point 10:A diagram of the acquisition of experiment data would be helpful to the reader for a better understanding of the test scenarios. Analogously, a diagram of the overall methodology would be helpful for the readers.
Reply :
Diagrams has been added in “2.3. Experimental Process” on page 5.
Point 11:Please consider rephrasing sentence of line 141 page 5 ([...] Acknowledge4.2 [...]).
Reply :
The sentence in line 167 on page 6 has been rewritten.
(The sEMG signals of the target muscles were collected by the MP150 multi-channel physiological signal acquisition system. Meanwhile, Acknowledge4.2 was used to record data.)
Point 12: More than 10 releases have been made of MATLAB since the release of MATLABv2016. This also applies to SPSS 19.0. The software utilised should be up to date unless the usage of older releases is justified. Moreover, please specify the tasks implemented by these softwares.
Reply :
Since the software on the school lab computer was the old version, the software used in the experiment was not the latest version. After using the latest version of the same operation, the experimental results doesn’t show much difference. So we updated the description at line 172, The experimental data were exported and analyzed by SPSS 26.0. MATLAB R2022b was used to program recognition models in the after research.
Point 13:In line 150 of page 5 the authors stated that “Based on existing research, [...]”. Please provide more information about the research considered by the authors to sustain such a fact. Analogously, the extraction of both PSE and WPE should be justified.
Reply :
There was a reference missing in the line 196 on page 6. Now we have corrected it and listed it as reference [38].
Point 14:The authors indicated in page 5 that “this experiment extracted the PSE and WPE form the sEMG signal and analyzed the correlation between these features and dynamic muscle fatigue”. However, there is no evidence that the correlation mentioned was performed within the manuscript. Please provide further information about this process and the results obtained.
Reply :
The corresponding process and results of PSE and WPE have been added in Section 2.4.2.
Take VM for example, the PSE results are shown in Figure 3. At the two speeds, the PSE showed a steady downward trend with the movement process. The PSE had a very significant negative correlation with the number of movement (P <0.01), and the correlation coefficient was r=0.869 (slow) and r=0.818 (fast). When moving at 60 °/s, the power spectrum entropy decreased slowly. Through linear fitting, the slope of the fitting curve was -0.0156. When moving at 180 °/s, the slope of the linear fitting curve was -0.0204.
Also take VM for example, the WPE results are shown in Figure 3. At the two speeds, the WPE also showed a steady downward trend with the movement process. The WPE had a very significant negative correlation with the number of movement, and the correlation coefficient was r=0.853 (slow) and r=0.829 (fast). When moving at 60 °/s, through linear fitting, the slope of the fitting curve was -0.0144. When moving at 180 °/s, the slope of the linear fitting curve was -0.0142.
It can be seen from the above results that, when doing isokinetic knee flexion and extension at two speeds, the PSE and WPE of VM showed a good regular decline with the deepening of fatigue, which had a good characterization of dynamic muscle fatigue, and can be used as the sEMG features to identify muscle fatigue during dynamic muscle contraction.
The data of other five muscles was processed with the same method, and the PSE and the WPE of the five muscles’ sEMG all showed significant negative correlation. The correlation coefficient is listed in Table 3. Therefore, the PSE and WPE of these six muscles can be used as the signal features of muscle fatigue identification in the isokinetic knee joint flexion and extension exercise. In the subsequent dynamic fatigue recognition, the PSE and WPE of these six muscles will be used as the fatigue features to build a fatigue recognition model.
Point 15: If the reviewer understood correctly, a total of six acquisition channels were considered. Each channel recorded a sEMG. Then PSE and WPE were extracted from the sEMG in order to assess muscle fatigue. How is this assessment performed? Why is this step requried? How is this related to the CNN methodology part? Section “2.4.1. sEMG signal processing and analysis” is confusing with the information currently presented in the manuscript.
Reply :
The process of this assessment has been added in the manuscript on page 7 and 8. The assessment was performed to verify the feasibility of the PSE and WPE in characterizing muscle fatigue so that they can be used as features in the classification of Multi-SVM and Multi-LDA. The CNN was trained with the initial sEMG signals instead of the PSE and WPE. Relevant statements have been modified accordingly in the article.
Point 16:Was there any challenge with regards to missing values and errors introduced by the experimenter when determining the muscle fatigue state and acquiring the sEMG signals? How did the authors address that?
Reply :
As described in lines 132-137,To reduce the impedance, it is vital to remove the body hair of the tested parts and wipe it with alcohol to remove the surface stains. After the alcohol was air-dried, the Ag-AgCl electrodes were pasted on and connected to the sEMG acquisition channels through wireless connection. Subsequently, the subjects randomly applied force, and the experimenter observed the collection of EMG signals and verified that the connection was feasible.
Point 17: A comparative study was performed to validate the performance of CNN. Specifically, the SVM and LDA models were considered. Please provide adequate justifications on the implementation of these models as part of the comparative study. Moreover, other state-of-the art models should be also considered to ensure credibility of the comparative study. For instance, how your model compares to the one proposed by Wang et al. (2021) (please see point 1 for full reference).
Reply :
In order to identify the initial fatigue status, our study divided fatigue into four categories and increased the number of subjects. More information is described in the reply to point 1.
Point 18: It seems that references are missing in line 222 of page 7 ([…] neurons []).
Reply :
As described at line 287, References [38] have been added.
Point 19:Please justify the selection of the architecture of the CNN and the shape of the inputs. For instance, why did the authors decide to use a unique full connection layer? Why were a total of 120 sampling points considered to define the input shape of the CNN? Did the location of the six target muscle channels in the input matrix (current definition: [SEM, BF, SE, REC, VL, VM]) influence in the accuracy of the model? The authors created a total of two CNN models: one for the 60º/s, and another for the 180º/s. Did they present the same architecture? Why?
Reply :
One fully connected layer is sufficient to meet the classification requirements of this experiment.
Within a total of 40 movements during 60 °/s training, samples were taken every 1s while every movement was 3s. A total of 120 sEMG signals had been sampled. Therefore, the input shape of the CNN was set to be 120 sampling points. As for 180 °/s, within a total of 40 movements, samples were taken every 250ms while every movement was 1s. A total of 160 sEMG signals had been sampled. So the input shape of the CNN was set to be 160 sampling points.
It is possible that different location of the six target muscle channels in the input matrix might influence the accuracy of the model. Before the experiment, we had designed 3 different locations of the six target muscle channels in the input matrix (a. [SEM, BF, SE, REC, VL, VM]; b. [SEM, SE, VL, REC, BF, VM]; c. [BF, REC, VL, SEM, VM, REC] ). After comparing the results of these 3 different locations, we found that there was little influence. whether other different locations would influence the the accuracy of the model remained to be studied.
Two CNN models for different speeds present the same architecture because the experiments were performed with the same method and the channels.
Point 20:The consideration of the confusion matrix as evaluation metric is recommended.
Reply :
The confusion matrix has been added in “3.2. Exercise fatigue recognition results based on test samples”.
Point 21:The reviewer considers that the results section needs to be expanded. For instance, more information about the results from the data acquisition and data pre-processing processes needs to be provided (e.g., visualisations of the signals, descriptive statistics, correlation analysis, etc.). Also, more information about the dataset needs to be provided. For instance, the number of samples per category and test. Is the dataset imbalanced? How did the authors address this issue?
Reply :
More results has been added in “2.4.2. sEMG signal processing and analysis”. The results of data acquisition and data preprocessing were omitted considering that they were not important in this study.
The number of samples per category and test are listed below:
Multi-SVM: training (36) : validation (12) : test (12)
Multi-LDA: training (36) : validation (12) : test (12)
CNN: training (36) : validation (12) : test (12)
The ratio “3:1:1” is widely used in the researches about classification models.
Point 22:Multi-SVM, and Multi-LDA were implemented as part of the comparative study. However, there is no evidence about the hyperparameter optimisation performed by the authors. For instance, how did the authors estimate the kernel, gamma, and C hyperparameters of multi-SVM?
Reply :
Information about Multi-SVM and Multi-LDA has been added in “2.4.4. Experimental sample construction”.
(1) Multi-SVM: Used PSE and WPE of 6 muscles’ sEMG as features, and then classified them by SVM classifier using Gaussian kernel function. The kernel function formula is as follows.
|
|
|
(19) |
(2) Multi-LDA: The sEMG’s PSE and WPE of 6 muscles were also extracted, and then classified by Multi-LDA classifier. Mark the characteristics of the input sEMG signal as xi (i=1, 2, 3,...., n), set the input sample set as , where corresponds to the LDA classification label as Xa (a=1,2,...,4). Based on the above marks, the four-classification algorithm formula of Multi-LDA's is as follows:
|
|
LDA()= |
(20) |
In the formula, SB represents the between-class scatter matrix, SW represents the within-class scatter matrix. The Multi-LDA is maximized to achieve the maximum SB and the minimum SW, so that the dimension-reduced classification sample obtains the maximum inter-class distance and the minimum intra-class distance.
Point 23:A table providing the results for each model and evaluation metric is recommended.
Reply :
The results for each model and evaluation metric has been listed in “Table 4. The Accuracy of the fatigue identification in 60°/s and 180°/s training by 3 models.”
Point 24:Four additional participants were considered to further verify the practical application of the model. Please provide the subject basic information of these participants.
Reply :
The relevant description has been corrected. As replied to point 5, we recruited 64 men of similar age and body size as subjects.
Point 25:The authors stated that after adding the additional participants to the training the accuracy increased to 92.16%. Please indicate if this 92.16% relates to the test set or to these four participants, as if these refers to the four participants there is a clear biased evaluation.
Reply :
92.16% is the recognition rate obtained after adding the rest four subjects’ data to the CNN training data, which is higher than the previous CNN model.
Point 26:Which are the disadvantages and limitations of the proposed model?
Reply :
The model was classic. After years of development, After years of development, there may be some other more advanced and effective models. Therefore, it is possible to further improve the recognition rate of dynamic muscle fatigue in knee joint isokinetic flexion and extension training. We will study it in the future.

Reviewer 2 Report
Title: Dynamic Muscle Fatigue Recognition Based on Deep Convolutional Neural Network
Submission ID: healthcare-1988257
This manuscript presented a research study that claimed to better identify the state of dynamic muscle fatigue in resistance training by investigating the isokinetic flexion and extension strength training of the knee joint and found that the CNN model has better classification performance as compared to two other models Multi-SVM and Multi-LDA. Almost this manuscript is well written and presented except few sentences require to be restructured.
Following are my comments to consider and address:
1. Line 93-94: “physical fitness of the selected participants was relatively close.”— How did researchers ensured it for all 60 participants? Did they follow any standard assessment test or criteria for it?
2. It is suggested that either change “Table 1” in to a sentence, or list more relevant information about the participants, like, the information used to assess their physical fitness.
3. Except Figure 2, all figures are in poor quality. Please use high resolution clear images and to insert, follow the journal’s guidelines.
4. Line 105, 117: “shown in Table2.”—a better word used to refer a table is “listed,” normally, shown is used with figures, please consider to correct such changes throughout the manuscript.
5. It is suggested that equations (expressions) to be numbered as required and use that number for giving reference in the different sections of the manuscript.
6. Line 215-217: “this paper also used SVM and linear discriminant analysis (LDA) to build classification models on the same training data and tested it on the same test data.”— This sentence is quite ambiguous.
If the same data is used to train and test, then there are high chances of data leakage (information leakage), how did authors ensure no data leakage.
If it means the same training and testing data as for the CNN model is used with SVM and LDA model, then please try to restructure the sentence for clear meaning.
7. Authors must define the different data-terms used in this manuscript to avoid any misunderstanding, such as in Line 270: “First, we input the preprocessed training data”—the term ‘preprocessed training data’ is quite ambiguous, it requires a little description about preprocessing sEMG signals and data split method and ratio used to split the data into training and testing datasets.
8. Line 330: “as shown in”— followed by an expression for accuracy, try to improve such sentences throughout the manuscript for better understanding. For a suggestion, it can be written as— ‘as or as given or can be expressed as,’ etc.
9. Line 344: “The training and validation data were used to train the CNN model.”— How come validation data is used to train the model? If it is done, then at the validation stage, model may suffer with data leakage. Authors must clarify this point and also discuss how data leakage was avoided.
As per the ML theory, a validation dataset is a sample of data held back from training the model which is used to give an estimate of model’s skill performance for tuning model’s hyperparameters. In other words, the validation dataset is different from the test dataset which is also held back from the training of the model, test dataset is used to give an unbiased estimate of the skill of the final tuned model when comparing or selecting between final models.
10. Line 387: “we selected four additional participants to perform isokinetic knee flexion and ex”— are these participants other than 60 participants mentioned in “Subjects,” section 2.1, line 92?
If yes, then please do mention about it in the section 2.1; if no, then how these 4 new participants were assured for having ‘relatively close’ physical fitness same as previous 60 participants?
11. Line 406-407: “CNN model was trained using the experimental data”— again, here a new term ‘experimental data’ is used which must be predefined. Also, Line 344 says, training and validation data were used to train the CNN model. Please explain.
Authors have to clarify about data-terms used, data split, and datasets used for training, validation and testing.
Overall, this manuscript is well written and presented but still lacks in expressing the clear meaning with respect to the different data-terms used and the datasets used for model’s training and validation process.
Author Response
Sincerely , we appreciate your valuable comments. About your questions, after careful thinking, we found that our research still has some deficiencies and had made corresponding improvements. The following will be the answers to your questions.
Point 1: Line 93-94: “physical fitness of the selected participants was relatively close.”— How did researchers ensured it for all 60 participants? Did they follow any standard assessment test or criteria for it?
Reply :
As described in lines 99-104, we recruited 64 men of similar age and body size as subjects. Reference [31] indicate that both gender and age factors can affect the subjective RPE. Considering the safety of this experiment, men of similar age were selected as subjects.
Point 2: It is suggested that either change “Table 1” in to a sentence, or list more relevant information about the participants, like, the information used to assess their physical fitness.
Reply :
As described in lines 99-100, In this experiment, 64 healthy men (n=64; age: 25.8±1.85years; Height: 174.72±3.88cm; Weight: 64.75±3.42kg) with similar body size were recruited as the participants.
Point 3:Except Figure 2, all figures are in poor quality. Please use high resolution clear images and to insert, follow the journal’s guidelines.
Reply :
The related images have been updated.
Point 4:Line 105, 117: “shown in Table2.”—a better word used to refer a table is “listed,” normally, shown is used with figures, please consider to correct such changes throughout the manuscript.
Reply :
The relevant words in the full text have been modified.
Point 5: It is suggested that equations (expressions) to be numbered as required and use that number for giving reference in the different sections of the manuscript.
Reply :
The equations in this article have been numbered.
Point 6:Line 215-217: “this paper also used SVM and linear discriminant analysis (LDA) to build classification models on the same training data and tested it on the same test data.”— This sentence is quite ambiguous.
If the same data is used to train and test, then there are high chances of data leakage (information leakage), how did authors ensure no data leakage.
If it means the same training and testing data as for the CNN model is used with SVM and LDA model, then please try to restructure the sentence for clear meaning.
Reply :
As described in lines 418-426, three models were trained with the same training data and tested with the same test data.
Point 7:Authors must define the different data-terms used in this manuscript to avoid any misunderstanding, such as in Line 270: “First, we input the preprocessed training data”—the term ‘preprocessed training data’ is quite ambiguous, it requires a little description about preprocessing sEMG signals and data split method and ratio used to split the data into training and testing datasets.
Reply :
The relevant statement has been modified in 2.4.1 and 2.4.3. The preprocessed data refers to sEMG data after being normalized using MVCs.
Point 8:Line 330: “as shown in”— followed by an expression for accuracy, try to improve such sentences throughout the manuscript for better understanding. For a suggestion, it can be written as— ‘as or as given or can be expressed as,’ etc.
Reply :
The relevant statements in the full text have been modified.
Point 9: Line 344: “The training and validation data were used to train the CNN model.”— How come validation data is used to train the model? If it is done, then at the validation stage, model may suffer with data leakage. Authors must clarify this point and also discuss how data leakage was avoided.
As per the ML theory, a validation dataset is a sample of data held back from training the model which is used to give an estimate of model’s skill performance for tuning model’s hyperparameters. In other words, the validation dataset is different from the test dataset which is also held back from the training of the model, test dataset is used to give an unbiased estimate of the skill of the final tuned model when comparing or selecting between final models.
Reply :
The relevant statements have been modified in 2.4.4 and 3.1.
There were 64 subjects in the experiment. Random 60 subjects’ data was selected out from 64 subjects. The experimental data of the 60 subjects were divided into 5 segments according to the ratio of 3:1:1. Three segments (60%) of the data were used as training data; One segment (20%) of the data was used as the validation data; One segment (20%) of the test data was used as the test data. The training data was used for constructing the model, while the validation data was used to select the optimal parameters of the model, and the test data was used to evaluate the model recognition rate. The data of the rest 4 subjects was used for further model validation and did not utilised in the training.
Point 10: Line 387: “we selected four additional participants to perform isokinetic knee flexion and ex”— are these participants other than 60 participants mentioned in “Subjects,” section 2.1, line 92?
If yes, then please do mention about it in the section 2.1; if no, then how these 4 new participants were assured for having ‘relatively close’ physical fitness same as previous 60 participants?
Reply :
The relevant statements have been modified in 2.1. 64 subjects were recruited at the same time. 4 of the subjects’ data was used for further research.
Point 11: Line 406-407: “CNN model was trained using the experimental data”— again, here a new term ‘experimental data’ is used which must be predefined. Also, Line 344 says, training and validation data were used to train the CNN model. Please explain.
Authors have to clarify about data-terms used, data split, and datasets used for training, validation and testing.
Overall, this manuscript is well written and presented but still lacks in expressing the clear meaning with respect to the different data-terms used and the datasets used for model’s training and validation process.
Reply :
The relevant statements have been modified in 2.4.4. The experimental data of 60 subjects were divided into 5 segments according to the ratio of 3:1:1. Three segments (60%) of the data were used as training data; One segment (20%) of the data was used as the validation data; One segment (20%) of the test data was used as the final test data. The training data is used to construct the model, the validation data is used to select the optimal parameters of the model, and the test data is used to evaluate the model recognition rate. The data of 4 subjects was used for further model validation and did not participate in the training.

Reviewer 3 Report
1. The authors have used 120 samples for NN, is that will be satisfy to predict the target in NN technique.
2. In the end of this manuscript the author has to list the steps of Matlab NN
3. The authors have to explain more details the own CNN, so explain the Figure 3.
4. The authors supposed to be used the FEM to comparison, So, the authors have used SVM & LDA and are both is not FEM therefore it is not satisfying your method. NN is a tool to predict and FEM is analysis.
5. This paper is not novel so remove any word like this.
6. The authors have to develop the introduction of this manuscript.
7. The authors have to put the numbering of each equation that used it.
8. sEMG, the authors have to explain that method with more details.
Author Response
Sincerely , we appreciate your valuable comments. About your questions, after careful thinking, we found that our research still has some deficiencies and had made corresponding improvements. The following will be the answers to your questions.
Point 1: The authors have used 120 samples for NN, is that will be satisfy to predict the target in NN technique?
Reply :
120 samples was satisfied to predict the target in NN technique.
Within a total of 40 movements during 60 °/s training, samples were taken every 1s while every movement was 3s. A total of 120 sEMG signals had been sampled. There was a total of 43200 (60*6*120) sEMG signal samples.
As for 180 °/s, within a total of 40 movements, samples were taken every 250ms while every movement was 1s. A total of 160 sEMG signals had been sampled. So There was a total of 57600 (60*6*160) sEMG signal samples.
Point 2: In the end of this manuscript the author has to list the steps of Matlab NN.
Reply :
The research in this paper came from the key technology research of major technology project in Zhejiang, China), involving the transformation of commercial projects. Therefore, the steps of Matlab NN are inconvenient to display. We would appreciate your understanding.
Point 3:The authors have to explain more details the own CNN, so explain the Figure 3.
Reply :
The relevant information for the CNN is displayed below Figure 5.
Point 4:The authors supposed to be used the FEM to comparison, So, the authors have used SVM & LDA and are both is not FEM therefore it is not satisfying your method. NN is a tool to predict and FEM is analysis.
Reply :
FEM is mainly used for mechanical analysis. However, this experiment only involves the classification and identification of muscle fatigue states, involving the classification algorithm. Although an isokinetic muscle strength training device was used in this study, maximum muscle strength was not thoroughly studied. So FEM was not used.
Point 5:This paper is not novel so remove any word like this.
Reply :
The relevant description in the full text has been modified
Point 6:The authors have to develop the introduction of this manuscript.
Reply :
The introduction section has been modified and refined
Point 7:The authors have to put the numbering of each equation that used it.
Reply :
The equations in this article have been numbered.
Point 8: sEMG, the authors have to explain that method with more details.
Reply :
As described in “2.3 Experimental Process” , to reduce the impedance, it is vital to remove the body hair of the tested parts and wipe it with alcohol to remove the surface stains. After the alcohol was air-dried, the Ag-AgCl electrodes were pasted on and connected to the sEMG acquisition channels through wireless connection. Subsequently, the subjects randomly applied force, and the experimenter observed the collection of EMG signals and verified that the connection was feasible.

Round 2
Reviewer 1 Report
The reviewer would like to thank the authors for the clarifications provided. The reviewer considers the manuscript can be accepted.
Author Response
Dear reviewer:
Thank you for your valuable comments.
According to your suggestions, we have made point-to-point modifications
We have uploaded the notes which is typed in the following Word.
Please open the file which is named “response to reviewer 1”for review.
Thank you very much!
Best regard to you!

Reviewer 3 Report
The manuscript is good to submit.
Author Response
Dear reviewer:
Thank you for your valuable comments.
According to your suggestions, we have made point-to-point modifications
We have uploaded the notes which is typed in the following Word.
Please open the file which is named “response to reviewer 3”for review.
Thank you very much!
Best regard to you!
